# The Motivation of Urban Gardens in Mountain Areas. The Case of South Tyrol

**Valentina Cattivelli**

Eurac Research, 39100 Bolzano/Bozen, Italy; Valentina.Cattivelli@eurac.edu or valentina.cattivelli13@gmail.com

**Abstract:** Urban gardens have attracted considerable academic attention in recent years. Several studies have, in fact, emphasized their positive contribution in terms of social integration, community health, urban regeneration, and food security, and explored individual gardeners´ motivations behind these practices. While these topics are well-documented with reference to metropolitan urban areas, few studies have been carried out in relation to other contexts such as mountain areas. This limited interest is probably due to the reduced urbanization of these areas, a preference for other forms of horticulture (essentially those practiced in people's own homes) or the use of different solutions to mitigate the negative effects of social problems. The recent proliferation of urban gardens in South Tyrol (IT) makes this mountain province an interesting laboratory for practices and narratives associated with socially innovative urban gardening experiences. This paper presents a characterization of all urban gardening initiatives in South Tyrol through cartographical representation. It explains gardeners´ and public institutions´ motivations, as well as non-gardeners' perceptions of urban gardening. Semi-structured interviews were conducted in the various South Tyrolean municipalities where urban gardening projects have been undertaken. The results suggest the great importance of the social and environmental aspect of urban gardens, and an interest in reconnecting with food practices even when food access is not a priority.

**Keywords:** urban gardens; urban gardening; South Tyrol; mountain areas; motivations

## 1. Introduction

In recent years, interest in urban gardens (UGs) has increased considerably [1]. Although the number of urban gardeners and data on characteristics of UGs on a global scale are largely unknown, the number of people involved—or who would like to be involved—in urban gardening activities at an urban level has grown steadily over the past few decades [2,3].

Urban gardens are generally defined as "open spaces which are managed and operated by members of the local community in which food or flowers are cultivated" [4]. As such, these gardens are similar to, but different from, backyard gardens, which are privately managed by families. As noted by Ferris, Norman, and Semplik, the two forms of garden differ in terms of "ownership, access, and degree of democratic control" [5]. This is particularly true of community gardens, which are collective and self-organized spaces cultivated by gardeners, under municipal regulation. Originally the product of grassroots movements, they have since assumed an official structuring and follow a joint political agenda based on solidarity and horizontal decision-making [6,7]. Other gardens that exist in addition to backyard gardens are intercultural gardens (emerging as a solution to satisfy the need of migrants and refugees), educational gardens (in schools, used to teach children about growing plants, etc.) therapeutic gardens (which supplement treatment for people with mental disorders), and private gardens (cultivated on land that private citizens rent out on terms they set) [8–10]. (Intercultural gardens, educational gardens, therapeutic gardens, and private gardens will not be analyzed in the present study).

The history of UGs begins with the Industrial Revolution in Europe. Families living in precarious economic conditions, suffering social exclusion and malnutrition, began cultivating the "gardens of

the poor" (in English, migrant gardens, in French, *jardins ouvriers*), set up on plots owned by local administrations, factories, or religious communities, with the aim of alleviating their difficulties in terms of food provision by cultivating vegetables and breeding small animals. The first association of individuals, families, or small communities dedicated to the cultivation of urban gardens was established in Germany in 1864. It was founded by a doctor named Schreber, with whom the idea of urban gardens is still so strongly associated that these horticultural initiatives are named after him: "Schrebergarten". In 1921, German urban gardens became part of *the Bundesverband der Gartenfreunde* (Federal association of friends of gardens), which today has about 1.5 million members in about 15,000 associations. In France, the history of urban gardens officially began in 1896, when the "Ligue du Coin de Terre et du Foyer" and "Ouvre de Jardins Ouvriers" were created to help families in serious economic difficulties. The UG movement spread rapidly during the First World War and, in 1926, under the stimulus of the French League, the Office lnternational du Coin de Terre et des Jardins Ouvriers was established by delegates from the gardens in France, Luxembourg, Austria, England, and Germany. The usefulness of urban gardens became even more important during the two World Wars, when the social and economic situation was challenging, especially in terms of food. During the 1970s and the 1980s, UGs developed mainly where there was a massive urbanization process, connected to mass immigration phenomena. The agricultural-productive function was regarded as the primary function in its own right. Urban gardens provided a product that served predominantly for own consumption, but that also served as an element of a barter economy. Since the 1990s, growing prosperity in industrialized countries has put garden productivity, understood as income support, in second place. The produce is still important, but other requirements have become more important, such as the possibility of having fresh food and knowing more about the production process [11].

Community gardens have always played an important role during times of crisis. Gardening exploded during the Second World War, and its success continued even after the end of the war. Its importance has increased with each economic crisis, from the Great Depression up to the recent 2008 financial crisis. This will probably also happen in this period of Covid-19. Gardening provides an immediate solution to the current challenges in terms of food access and provision of emergency food relief through local and diversified food practices, which are environmentally friendly and contribute to local and community food self-sufficiency.

Many international agencies and institutions have included UGs in their policies and programs of activities. For example, the European Commission includes UG analysis among its research and educational purposes, as described in its study on urban and peri-urban agriculture in the EU [12]. The signatories of the Milan Urban Food Policy note the importance of UGs as regards increasing social and economic equity, and invite local policy makers to "promote networks and support grassroots activities (such as community gardens, community food kitchens, etc.) that create social inclusion and provide food to marginalized individuals" (p.4). In the same document, they note the importance of UGs in terms of land protection and recommend them as a solution to protect and enable secure access and tenure to land [13].

Scientific literature explains the interest in UGs based on their different functions, ranging from social benefits and effects on the wellbeing and health of local communities, to food security and environmental preservation [14,15]. Specifically, it explores the contribution of UGs to reducing social isolation, their positive impacts on the psychological condition, particularly among older people, and occasionally their influence on familial and working relations [16]. Literature also investigates their contribution to lowering human pressure on natural resources by reducing pollution and mitigating environmental alterations [17]. Finally, it places equal focus on the role of UGs in increasing access to, and quality and affordability of, local food, highlighting their positive effects [18].

These functions have been observed in several cities around the world, including New York [19], Milan [20], and Barcelona [21]. Functions related to food security are particularly well-documented in the American metropolitan areas [18,22], and are used specifically to study UGs in disadvantaged neighborhoods and their role in building good citizenship. In France, scientific debate focuses on the

multifunctional nature of UGs and tends to focus on social and food accessibility questions, as UGs are regarded as laboratories for social and urban agriculture [23]. In Spain and Italy, recent scientific works consider the social and environmental aspects, and promote discussion on solutions to include UGs in town planning [24,25].

Recent studies also offer extensive information on gardeners´ individual motivations. For example, Draper and Freeman [26] review and analyze recent publications on collective gardens in the USA, and reveal eleven kinds of motivations that underpin these practices (health benefits, food source, economic development, youth education, development and employment, use and preservation of open space, crime prevention, neighborhood beautification, leisure and outdoor recreation, cultural preservation and expression, social interactions, community organizing). Scheromm [27] lists 12 categories of motivations that drive the gardeners of Montpellier to cultivate their UGs, and notes a particular interest in reconnecting with farming, even when food production is not a priority. Ruggeri et al. (2016) [28] group together all motivations emerging from a study on gardeners in Milan into just seven categories, highlighting the importance of social aspects in particular.

Yet, though the literature provides important insights into UGs, some themes remain largely unexplored. One such theme is their existence in other territories (outside of urbanized areas). UGs have been extensively studied in urban and metropolitan areas, but little study has been done into their popularity in mountain areas. The lack of interest in these areas and their experiences of gardening is probably due to the limited number of UGs or their presence in just a few municipalities. This, in turn, may be attributed to the fact that people in these areas cultivate their own gardens and have no interest in other forms of gardening, or to the perception that the existing green areas are sufficiently large to mitigate human pressure. Nevertheless, mountain areas face similar pressures to more urbanized areas in terms of soil consumption, climate adaptation, and aging. It is probable that UGs could make mountain areas less vulnerable to these changes and represent an innovative example of social service provision.

Another understudied theme is the perspective of other actors involved in urban gardening initiatives. Indeed, gardeners are not the only protagonists in these practices. Local authorities/municipalities, enterprises, citizens, associations, and horticultural facilitators, among others, are also involved; yet the literature generally assigns functions to the gardens without clarifying whose perspective is being presented. It has been shown, however, that the functions and importance attributed to these practices can vary considerably depending on the speaker [29]. There is also little research into the perception of the role of urban gardens among those who live in municipalities that have carried out horticultural projects, but who are not directly involved as they did not rent a plot of land.

South Tyrol, in the North of Italy, is taken as a test area. This mountain province has clear geographical specificities (e.g., remoteness, morphological features, etc.) and some objective factors of disadvantage (e.g., depopulation throughout the region, with the exception of some urban centers, increased human pressure on natural resources, etc.). For the past two years, it has realized a moderate number of urban gardens in some municipalities, becoming one of the few provinces in the Alpine area to promote them.

The novelty of this paper is precisely that it seeks to fill this gap in knowledge by dealing with urban gardens in these understudied areas and considering the points of view of gardeners, municipal administration, and the wider population who are not directly involved.

This paper has two aims. The first is to present an original, up-to-date description of the recent urban gardening initiatives in South Tyrol. The relative information has been obtained by processing interviews carried out with the municipal offices. We intend to map out the spatial distribution of UGs in the province under examination, and list their main characteristics. The second aim of the paper is to discuss the various meanings attributed to gardening by mayors/municipal offices, gardeners, and non-gardeners. Municipalities and gardeners provide an insight into their motivations, i.e., the expected benefits and functions they assign to UGs in interviews in terms of social integration, health implications, food security, and environmental effects. Some people living in municipalities

with urban gardens who are not directly involved in these projects have also been interviewed: Their opinions offer an equally important insight into what UGs represent for them, and the importance they attribute to these cultivation practices.

Collecting all of these opinions on UG cultivation experiences helps municipalities restructure and improve UG management, to meet the interests of gardeners and the local population and to maximize participation. This exercise is also useful in academic terms, as it offers new information on motivations surrounding UGs, in addition to the well-known driving factors expressed by gardeners who cultivate in urban and metropolitan areas, and by non-gardeners and municipalities operating in those areas.

The article is structured as follows. The second paragraph profiles motivations surrounding urban gardening, as emerge from recent studies. The third paragraph details the possible effects of urban gardening in mountain areas, as well as the importance of analyzing the points of view of all actors involved in these initiatives. The next sections describe the method adopted in the present study. This is followed by a description of the social and territorial characteristics of South Tyrol, the test area. The subsequent section outlines the current situation of urban gardening in South Tyrolean municipalities. The reasons for the diffusion of UGs in these municipalities are then presented, together with an exploration of the various meanings behind them. The final section outlines the conclusions drawn.

## 2. Profiling the Motivations behind Urban Gardens

A review of the existing literature on UGs reveals that the motivations for people to get involved in gardening are diverse. In this regard, Ruggeri et al. (2016) refer to the "complex mosaic of interests, possibilities, expectations, cultures, values and traditions of each individual" [28]. Numerous scholars have tried to identify these and group them into categories (e.g., Schromm, 2015). The present analysis applies the classification model developed by Cattivelli [20], who identifies four categories: Social integration, community health, urban regeneration, and food security.

The first category includes motivations linked to the role of UGs in promoting social integration [30]. Serving as a more relaxed gathering place than shopping malls or leisure centers, for example, UGs facilitate the deepening of social ties among gardeners, enabling them to build new social relationships outside of familial and work settings [31,32]. This in turn helps combat loneliness, a common condition affecting approximately one in three adults in urban areas, and particularly older people [33]. This condition depends on the familiar choices: An increasing number of single-family households [34], a decline in social relations due to the intensity of work [35], or for aging reasons [36]. Older generations have fewer social supports due to changes associated with their life stage (retirement, for example), age-related losses, and critical events (e.g., the marriage of sons or death of a partner). Victims of this social isolation, they spend a lot of time in their own houses and are more prone to developing conditions associated with loneliness (anxiety, depression, etc.) [37]. Families also regard gardening initiatives as "glue" that binds the relationships among their members. Spending more time together, family ties are strengthened: After a day of work or school, parents teach their children to cultivate or harvest fruits and vegetables, and grandparents educate their grandchildren in the practices of urban agriculture, but also in the principles of a healthier diet and respect for the environment [38]. There are also benefits to social relationships outside of the gardens, as Draper and Freedman (2010) [26] report. "I have met many people I didn't know and I can use this networking to get more things done in the town", explained one garden volunteer in a study by Ohmer et al. (2009: 391) [39].

Behind some social gardeners´ reasons is the belief that UGs are an expression of both personal and social identity. Their cultivation is expected to contribute to a person's sense of distinctiveness, self-esteem, self-efficacy, and continuity [40]. Moreover, it reflects important values and experiences with nature, as well as interaction with other people [41], offering fertile ground for nurturing a personal identity as a gardener and environmentally- and socially-connected person. Through cultivation, gardeners form attachments to a place and establish links between their sense of self and their home-spaces, creating and shaping individual and collective identities. As such, personal and social

identities promote social cohesion at a local level [42,43] and act as "glue" for the existing social capital that forms the basis of any kind of participation (politics, volunteering, etc.) [44–46]. This happens by fostering socialization among individuals from different educational and social backgrounds [47], or from different generations [48]. A positive link has been identified between social involvement and garden participation [49]: Gardeners participate more in local and neighborhood activities and public education meetings than non-gardeners. People involved in urban horticulture projects are those who have a greater perception of social capital than people not active in these projects [50].

In turn, the personal and social commitment to UGs produce a new repertoire of rituals and practices that do not destabilize the dominant agricultural and social culture, but rather enrich it. In fact, by exchanging information and learning new concepts, people learn and create new social models characterized by a rich variety of lifestyles, ethnicities, and age groups [51,52]. At the same time, they pass on popular anecdotes and local beliefs, by contributing to the preservation and transmission of peasant values and traditions that would otherwise be forgotten. This "preservation function" transforms UGs into incubators of social-ecological memory [53], which can be accessed and transferred to other people in different areas thanks to the interaction among gardeners during the cultivation process. Failure to protect such knowledge could result in a collective 'forgetting' of important social and ecological information and reduce local social and ecological resilience.

The second cluster of motivations includes gardeners´ aspirations to improve their own health and that of their communities through cultivation [54]. They are aware that connection between humans and nature increases individual wellbeing and improves quality of life and human health [55]. Gardeners know that cultivation also has positive effects on hand and body strength and flexibility [56], and contributes to reducing bodily pain and a sense of fatigue and anxiety [57]. They are also attracted to gardening in order to experience joy and satisfaction through growing their own vegetables, to be outdoors in the natural environment, and to relieve stress [58,59]. Finally, they are motivated by a desire to enhance positive dietary habits [60] through increased consumption of fruit and vegetables and the option to consume whole food, freshly harvested, of certain origin and that has not been treated with fertilizers or polluting substances.

The third cluster of motivations includes those related to urban regeneration. As motivations are not decoupled from societal trends, "the increasing concerns for urban sustainability and greener and more inclusive cities also give shape to many of the reasons to be involved in urban gardening" [6] (p.:336). Among these motivations, the gardeners' belief that cultivation can affect the quality of life in their neighborhoods—by cleaning up vacant lots, reducing crime, or simply beautifying the neighborhood—is therefore considered relevant [60]. The same applies to the desire to support conservation of green spaces at risk of cementization [39]. UGs are in direct competition with other more profitable types of usage such as parking lots, residential building sites, etc. As such, there need to be motivations beyond such economic considerations, creating higher added value that surpasses economic interests. Moreover, the comparison between the environmental benefits and the costs of implementing and cultivating UGs remains largely uncertain. In terms of $CO_2$ production, the impact of UGs is negligible, and lower than other forms of cultivation. [61].

Motivations surrounding food security constitute the final cluster of motivations [62]. UGs improve food access and minimize transportation [63], as well as offering people the opportunity to turn their backs on industrial agriculture and adopt other methods of production [64]. Nevertheless, urban gardening does not automatically go hand in hand with cost saving [65] and, in certain places, food security and savings do not appear to be a prior motivating factor for gardeners (e.g., in Montreal) [66].

## 3. Different Points of View: Beyond Gardeners and Urban Areas

Despite the increasing focus on UGs, the presence in mountain areas has been little studied. The lack of interest is the result of the limited number and low diffusion of these experiences of gardening. Local populations prefer to cultivate their backyard gardens, which serve an utilitarian purpose, and have no particular interest in other cultivation initiatives. The existing green areas widely

available as part of the local landscape are considered sufficient to mitigate human pressure. Vacant public areas that might be allocated to collective cultivation are limited. Nevertheless, UGs could serve as an interesting laboratory for actions aimed at mitigating the current social and environmental pressures affecting these regions.

Many mountain areas are suffering from depopulation due to a deficit of births relative to deaths, negative net migration, or both. Depopulation also results from limited local access to job markets and public services, which reduces the attractiveness of these places and encourages out-migration. This, in turn, influences service provision, including food supply, which may be limited as a result. These territories are also facing population aging, caused by a decrease in births and prolonged increase in life expectancy [67]. As attractive tourism hotspots, mountain areas are also under pressure to build new links among local tourist areas, giving rise to increasing peri-urbanization [68]. Reducing transport links or discouraging individual car travel could, however, further limit access to these sites, thus decreasing the economic and social attractiveness of the areas [69].

Moreover, mountain areas are increasingly affected by climate change [70] and problems associated with ecological vulnerability [71,72]. This has direct implications for agriculture, the exploitation of the local natural ecosystem for tourist purposes, and on the quality and quantity of natural capital [73].

UGs could play a significant role in these regions and counteract the negative affects of the aforementioned social and environmental pressures.

They could provide a space for elderly people and families to enjoy themselves, interact with other people, and perhaps create a network of relationships among peers. This could reduce the risk of social exclusion and loss of local social capital, representing a first step toward bottom-up social innovation initiatives, which could then be launched to partially remedy the lack of public services. The creation of UGs could have effects on the environment and its quality, reducing the risk of vacant land being cemented and beautifying the landscape. Their cultivation could increase food availability, particularly in relation to vegetables. This could be important in places at risk of food desertification due to the closure of local shops or food distribution centers, and those poorly connected to the local transport infrastructure.

Although reasons may be subjective and heavily influenced by the local context in which the gardens are established, only a small number of studies investigate the motivations of gardeners in mountain areas.

Among these research projects, Ryan (2017) [74] reveals that in the mountain areas of Virginia (USA), UGs contribute to food sovereignty and community health, but have less of an impact on social cohesion, as local gardeners view themselves as an apolitical movement, removed from the more political goals of the organized food sovereignty movement. Engle [75] demonstrates that everyday environmental injustices in Central Appalachia place constraints on program participation and the beneficial outcomes of urban gardening programs, particularly for more disadvantaged households affected by chronic illness, geographic isolation, and environmental hazards. Although the interviewees regard these challenges as further justification for pursuing the program, these constraints were also found to interact in a way that limited the success of these locally-oriented sustainable development efforts, particularly for marginalized individuals. The majority of studies, however, focus on private gardens, which are more numerous (e.g., Reyes-Garcia (2010) [76].

Literature generally refers to "gardeners", omitting any in-depth descriptions of the perceptions of other individuals and institutions that collaborate to realize and manage urban gardening projects (local authorities/municipalities, enterprises, citizens, associations, horticultural facilitators, etc.). It has, however, been shown that the functions and importance attributed to such projects by different speakers can vary considerably. Duchemin et al. (2010) [77] indicate certain differences in responses, depending on the interviewee (gardeners, municipal offices, horticultural facilitators, etc.). In their studies focusing on Montreal, for example, they reveal a greater focus on food security and food-cost savings by local municipal offices, while these reasons were not a priority for local gardeners. In contrast, social interaction and contact with nature emerged as important motivations for the latter [78]. In another study on educational programs involving UGs, d´Abundo and Carden (2008) [54] outline a sort of asymmetry

between the motivations of schools and gardeners. The school administrators' goal of obesity reduction differed from the participants' motivation, which focused on wellness and community development.

## 4. Methods

This paper aims to present an original, up-to-date description of the recent urban gardening initiatives in South Tyrol, as well as discussing the different meanings of these experiences as perceived directly by municipal administrations, local gardeners, and the local population not directly involved in these initiatives.

In practical terms, all 116 South Tyrolean municipalities were contacted by email between July and October 2019, to inquire about the presence of UGs in their areas. Sixty-six municipalities responded. Of these, 58 municipalities stated they do not have any UGs, while the remaining 8 municipalities (Bolzano, Vipiteno, Vadena, Egna, Parcines, Salorno, Silandro, and Terlano) reported the presence of UGs. Fifty municipalities did not respond. Based on this preliminary information, a map of spatial distribution of UGs across South Tyrolean municipalities was created using Arcmap (GIS). An interview was carried out with mayors or municipal offices that promote UGs to get more information on the characteristics of the gardens. The first part of the interview includes some closed questions, the answers to which served to better explore the size and characteristics of the gardens, their management, and the profile of the gardeners (age, sex, and level of personal income). The second section featured just one open question, and a list of motivations. This open question concerned the motivations that drive local municipalities to promote urban gardening. Mayors, or municipal offices answering on behalf of the latter, were free to express all motivations underpinning their decision to promote these practices at a municipal level. Subsequently, they were asked to score the listed motivations on a scale of 0–5, based on their feelings (0 not at all important; 5 very important). The list includes all motivations referred to in the literature, grouped into 4 categories. In a similar study, Scheromm (2015) [27] considered 12 categories, whereas Ruggeri et al. (2016) [28] addressed just seven. In the present study, the 14 different themes that emerge from recent literature and are referred to in the previous paragraphs are clustered into the four categories identified by Cattivelli [20]: Social integration, health effects, urban requalification, and food security. Table 1 lists the meanings and their inclusion in the four mentioned categories:

The sentences used by mayors, or municipal offices responding on behalf of the latter, were processed using QCMap, free software for text and content analysis. Specifically, a directed content analyis was performed. The study started with theory analyisis, while codes were defined before data analyisis and were associated to the identified four categories (social integration, health effects, urban requalification, food security). Sentences were then re-organized and grouped into the four categories. The mayors´ scores were recorded in an excel sheet and presented on a radar graph.

Five of the eight mayors/municipal offices that responded declared that they were directly involved in realizing the community gardens in their municipal areas. The remaining three were not directly involved in the management of UGs, but had in the past facilitated their realization by amending the urban plans to rezone certain sections of private land. As such, the mayors/public offices did not speak for private citizens who loan their lands for gardening purposes. Unfortunately, these citizens did not respond to the interviews.

The second part of the interview was sent to the gardeners to investigate the specific meanings of gardening attributed by these actors to their gardening experience. Potential recipients of the questionnaires were identified mainly thanks to the cooperation of the mayors and municipal offices, and were contacted based on their strong record of attendance and the opportunity to meet with them directly at the gardens. Gardeners were contacted by phone or in-person at the gardens to explain the purpose of the research and schedule a personal meeting. Using this approach, 16 gardeners were interviewed. These gardeners cultivate community gardens, having rented them directly from the five municipalities. Gardeners who rented garden space from private owners were not interviewed, as relevant information was unavailable for privacy reasons.

**Table 1.** List of meanings considered in the present study and their relevant categories.

| Categories | Motivations | References |
|---|---|---|
| Social integration | Offering opportunities for socialization | [30] Mudu, P.; Marini, A. (2018)<br>[31] Hawkins, J.L.; Mercer, J.; Thirlaway, K.J.; Clayton, D.A. (2013)<br>[32] Glover, T. D.; Parry, D. C.; Shinew, K. J. (2005) |
| | Social support for elderly people | [33] Remes, O. (2018)<br>[34] ASTAT (2017)<br>[35] Yougov (2018)<br>[36] ISTAT (2018) |
| | Strengthening family ties | [38] Doyle, R.; Krasny, M. (2003) |
| | Social ecological memory of ancient practices | [51] Fusco, D. (2001)<br>[52] Lautenschlager, L.; Smith, C. (2007)<br>[53] Barthel, S.; Parker, J.; Folke, C.; Colding, J. (2014) |
| | Strengthening personal and social identity | [40] Crompton, T.; Kasser, T.; (2009) [41] Clayton, S. (2007) |
| | Strengthening social cohesion and participation | [42] Wang, D.; MacMillan, T. (2013)<br>[43] Veen, E.J.; Bock, B.B.; Van den Berg, W.; Visser, A.J.; Wiskerke, J.S. (2016)<br>[44] Glover, T. D. (2004)<br>[45] Henderson, B. R.; Hartsfield, K. (2009) |
| | Sharing value | [40] Crompton, T.; Kasser, T.; (2009) |
| Health effects | Increase the psycho-physical well-being of citizens | [54] D'Abundo, M.; Carden, A. (2008)<br>[55] Claessens, J.; Schram-Bijkerk, D.; Dirven-van Breemen, L.; Otte, P.; van Vijnen, H. (2014)<br>[56] Park, S.A.; Shoemaker, C.A.; Haub, M.D. (2009)<br>[57] Hartig, T.; Mitchell, R.; de Vries, S.; Frumkin, H. (2014)<br>[58] Van Den Berg, A.E.; Custers, M.H. (2011) |
| | Better food habits | [59] McCormack, L. A.; Laska, M. N.; Larson, N. I.; Story, M. (2010) |
| Urban requalification | Cleaning up vacant lots, reducing crime, or just beautifying the neighborhood | [60] Wunder, S. (2013)<br>[39] Ohmer, M.L.; Meadowcroft, P.; Freed, K.; Lewis, E. (2009) |
| | Protection of the environment and landscape | [39] Ohmer, M.L.; Meadowcroft, P.; Freed, K.; Lewis, E. (2009) |
| | Reducing pollution | [61] Favoino, E.; Hogg, D. (2008) |
| Food security | Access to food | [62] Opitz, I.; Berges, R.; Piorr, A.; Krikser, T. (2016)<br>[63] Armstrong, D. (2000) |
| | Turning back on industrial agriculture and search of other methods of consumption, but also production methods | [64] Mok, H.; Williamson, V.G.; Grove, J.R.; Burry, K.; Barker, S.F.; Hamilton, A.J. (2014) |
| | Access to food at cheaper conditions | [65] Cattivelli V.; (2016) |

Source: Own elaboration based on literature review, 2020.

All interviews were recorded and transcribed. Additionally, in order to investigate their motivations, participants were asked to give a score to all motivations based on their importance, from 0 (not at all important) to 5 (very important). These statements were grouped into the four categories and processed using QCMap. The gardeners´ scores were recorded in an excel sheet and an average was calculated for each category. These average values were then presented on a radar graph.

The same interview text was also presented to residents of the five municipalities where community gardens were realized. This is because one aim of this article is to investigate what UGs represent to those not directly involved in their cultivation. 20 people were identified, 10 of whom were from the older generation, and 10 of whom represent families. The first group was selected because older people are the main target of UG gardening practices, as highlighted by a significant portion of the literature. Families, meanwhile, are a "shadow" target group, with studies suggesting that they would like to rent gardens, but are generally excluded from allocation procedures [17]. These groups were asked about their knowledge and frequentation of UGs in their municipalities, and invited to respond to the same interview provided to the gardeners. They were contacted directly. They were asked to specify the meanings they attributed to UG practices and assign a score between 0 and 5 to the meanings listed in the table. The sentences were grouped and processed using the same software, and scores were calculated and recorded using the same methods applied to the gardeners' interviews.

## 5. The Test Area: South Tyrol

South Tyrol is an autonomous province and a largely mountainous area in the North of Italy [78,79]. This province is currently facing population aging, primarily due to an increase in life expectancy,

decrease in births [34], and the depopulation of small centers due to migration flows towards the largest local towns. Five-hundred twenty-four thousand, two-hundred and fifty-six people live in the region, 106,951 of whom live in its biggest city: Bolzano (ibid.). Approximately 40% of the population live in urban areas, while the rest is distributed across smaller towns in the valleys and mountains (ibid.). Figure 1 shows the mountainous nature of the province and illustrates the distribution of municipalities, identifying those with more than 5000 inhabitants.

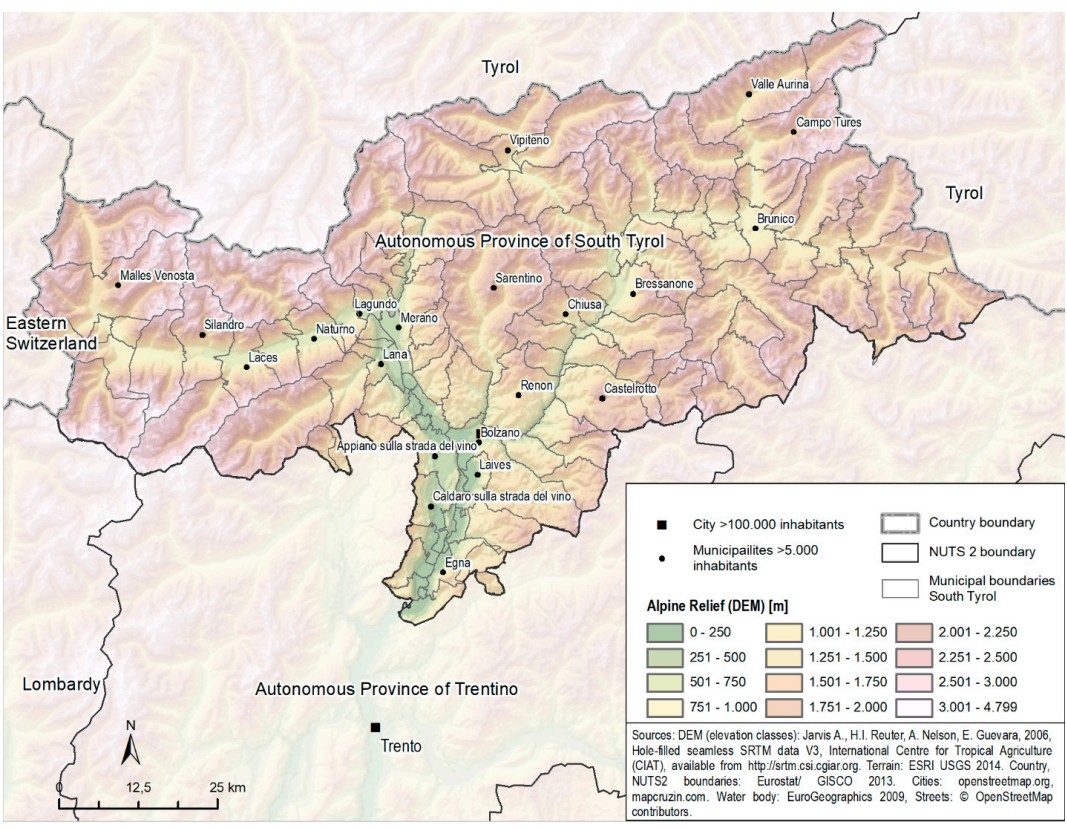

**Figure 1.** Mountainous nature of South Tyrol and the distribution of its municipalities.

The number of people over 60 and living alone is constantly growing. Social isolation and loneliness expose senior citizens to additional age-related risks, including illness, loss of family ties and friendship, difficulty asking for help, and reduced mobility. Those most exposed to these risks are the elderly living alone in major urban centers and those living in the most remote and smallest centers. Elderly people living in urban areas have trouble accessing support services due to their state of health or because they are not firmly integrated into the social life of their neighborhood. Residents in the most remote and smallest centers, meanwhile, struggle to access support services because of their distance from the larger centers. In the past, these small, remote, rural communities looked after their members through voluntary forms of mutual solidarity beyond family ties. In this way, local communities replaced certain support services with voluntary and free welfare initiatives. Such mutual support networks are now more difficult to maintain, and the system is likely to die out, due in part to generational turnover and associated problems with transmitting the values of solidarity to the younger generations [80].

All of these factors mean that the social structure of South Tyrol is changing, and its social capital may be at risk due to depopulation, aging, and decline in social relations.

South Tyrol has a remarkably varied landscape due to its morphological and climatic features. It is a geologically heterogeneous region, with silicate dominating in the west and lime in the east, and a range of altitudes including sub-Mediterranean zones in the valley and alpine zones in the mountains. As illustrated in Figure 2, high nature value (HNV) farmland, nationally designated areas

(CDD), Natura 2000 sites feature extensively throughout the province. The high incidence of these sites has led to South Tyrol being labelled a green province, protecting against the risk of depletion and urbanization due to demographic and climate factors. Various local authorities and stakeholders are involved in designating and managing the local natural landscape.

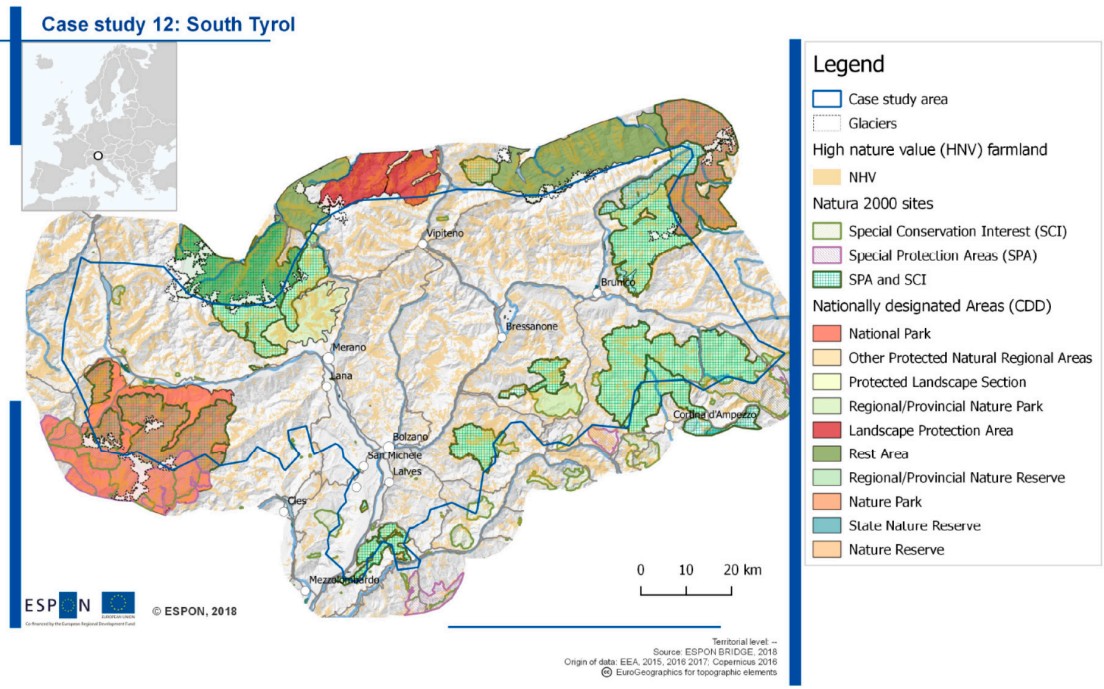

**Figure 2.** Map of high nature value (HNV), Natura 2000 sites, and nationally designated areas (CDD). Sources: In the map.

## 6. Urban gardens in South Tyrol

### 6.1. Characterization and Evolution of Urban Gardening Initiatives

Urban gardens are present in eight municipalities: Bolzano, Parcines, Egna, Salorno, Silandro, Terlano, Vadena, and Vipiteno. The number of such gardens varies greatly across these municipalities. Bolzano, the provincial capital, features 162 gardens, while the smaller municipalities such as Egna and Salorno have approximately 20–30 gardens. The mapped gardens are located in municipalities at low altitudes. The majority (5) are located at a height of less than 500 m above sea level. Only three municipalities feature gardens at a height of more than 600 m above sea level, with none located above 1000 m.

Figure 3 suggests their distribution is highly uneven and mainly concentrated in the largest, least remote municipalities, at low altitudes.

(The municipalities of Silandro and Parcines indicate that UGs are limited to a single area, but do not specify their exact number).

The size of the gardens varies greatly. Most allotments span 50 square meters. Others are smaller, at 30–40 square meters.

The relevant lands are mostly owned by the municipalities in Bolzano, Egna, Salorno, and Vadena (community gardens). Private citizens rarely make their own plots available free of charge or at low rents to others (this only happens in Terlano and Silandro), and can only do so following rezoning on the municipal plans (from arable land to urban garden). In Vipiteno, land is lent to residents for gardening purposes by both the local municipality and private citizens.

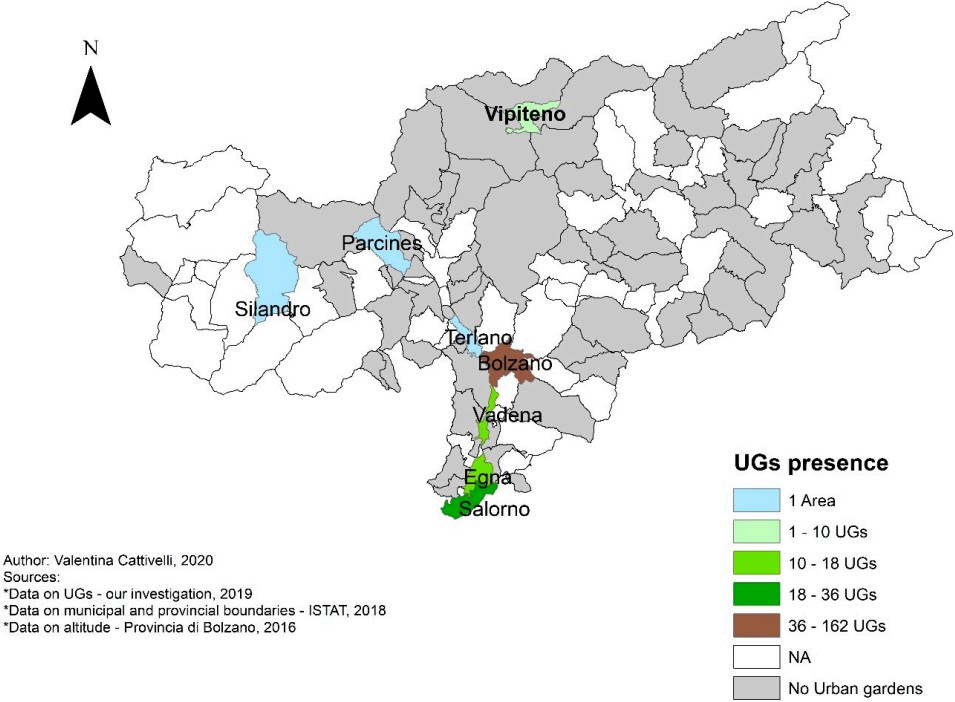

**Figure 3.** The distribution of urban gardens in South Tyrolean municipalities. Source: ISTAT, 2018 [81]; own investigation, 2019, Cattivelli, 2020 [82].

Through a public procurement process, local administrations rent the community gardens to the "aspiring farmers", who must meet certain requirements set out in municipal regulations. Usually, these relate to age (over 55 years), residence status (in the municipality for some years), or income (no/low income is one of the preferred conditions). In Vadena, families with children under 14 years of age can also access the selection process.

Gardens cannot be assigned to anyone who owns agricultural land and/or has the use of other land suitable for cultivation in the province of Bolzano. Normally, gardens are assigned for 5 years. Concessions may be renewed, but renewals will not be granted to those who previously abandoned their plot, or failed to properly tend it. The gardener cannot give it away or sublet garden, but must cultivate it directly and continuously. Assignees undertake to contribute to maintenance of the common areas in accordance with the Mayor's instructions and are required to help cover irrigation costs and charges relating to the garden. Gardeners cannot raise animals, sell produce from the garden, or damage the municipal facilities provided.

Currently, gardens in the municipalities referred to above are grown by individuals. The exception to this is in Bolzano, which includes a garden managed by a private association. The "typical" gardener is male, over 55 years of age, from South Tyrol, with a low–medium income.

In the case of urban gardens on private land, the parties involved are free to negotiate the rules for their use.

Both experiences are relatively recent: All municipalities and private individuals have been carrying out urban gardening projects for less than two years.

In some (rare) cases, lands are transformed spontaneously by citizens without permission, such as in Bolzano, where some squatted gardens have been identified. Bolzano also appoints a gardeners' representative, who acts as an intermediary between the other gardeners and the municipal administration.

Table 2 summarizes general information about UGs in the respondent municipalities.

**Table 2.** Some general information about urban gardens (UGs) in each South Tyrolean municipality, where present.

| General Information | EGNA | SALORNO | BOLZANO | VIPITENO | VADENA | PARCINES | SILANDRO | TERLANO |
|---|---|---|---|---|---|---|---|---|
| Number of gardens | 18 | 36 | 162 | 10 (possibility of increasing this number if there is demand) | 15 | 1 area | 1 area | 1 area |
| Average size | 33 m$^2$ | 40 m$^2$ | 40–50 m$^2$ | 10 m$^2$ | 35 m$^2$ | N/A | N/A | N/A |
| Altitude (m) | 214 | 224 | 262 | 948 | 243 | 626 | 721 | 248 |
| Users/Gardeners | Elderly people | Elderly people and the disabled | Elderly people, volunteer associations | Citizens of various age groups and social categories | Pensioners and families | Private individuals | Private individuals | Private individuals |
| Rental arrangements | Criteria-based selection | N/D | Criteria-based selection | Assigning a plot in exchange for a small contribution | Criteria based selection | Private agreement | Private agreement | Private agreement |
| Cost to gardeners | EUR 40/year | EUR 40/year | EUR 38/year | Approximately EUR 40.00 per year for water | EUR 60 per year and a variable contribution for general expenses | private agreement | private agreement | private agreement |
| Cost to local municipality (type) | Land rental | Extraordinary maintenance | Extraordinary maintenance | grass cutting, provision of a fence and tool shed | Extraordinary maintenance | Zero | Zero | Zero |

Source: Own elaboration based on direct interviews, 2019.

Figures 4–6 illustrate UGs in Bozen, Salorno, and Terlano, respectively. In Bozen, the capital city and the only urban center with more than 100,000 inhabitants, UGs are established in vacant spaces among residential buildings. In Salorno, a small, mountainous center, UGs are established in open areas near flat plots among the mountainous areas. In Terlano, UGs are cultivated on private land and rented by private individuals.

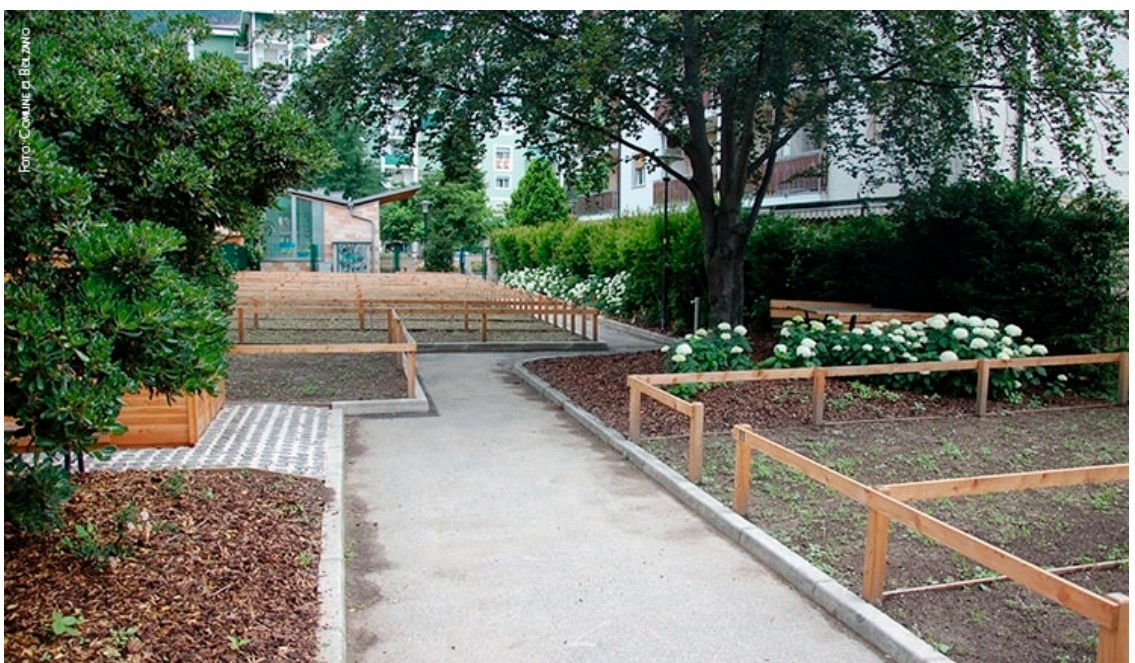

**Figure 4.** Some UGs in Bozen. Source: Municipality of Bozen, 2020.

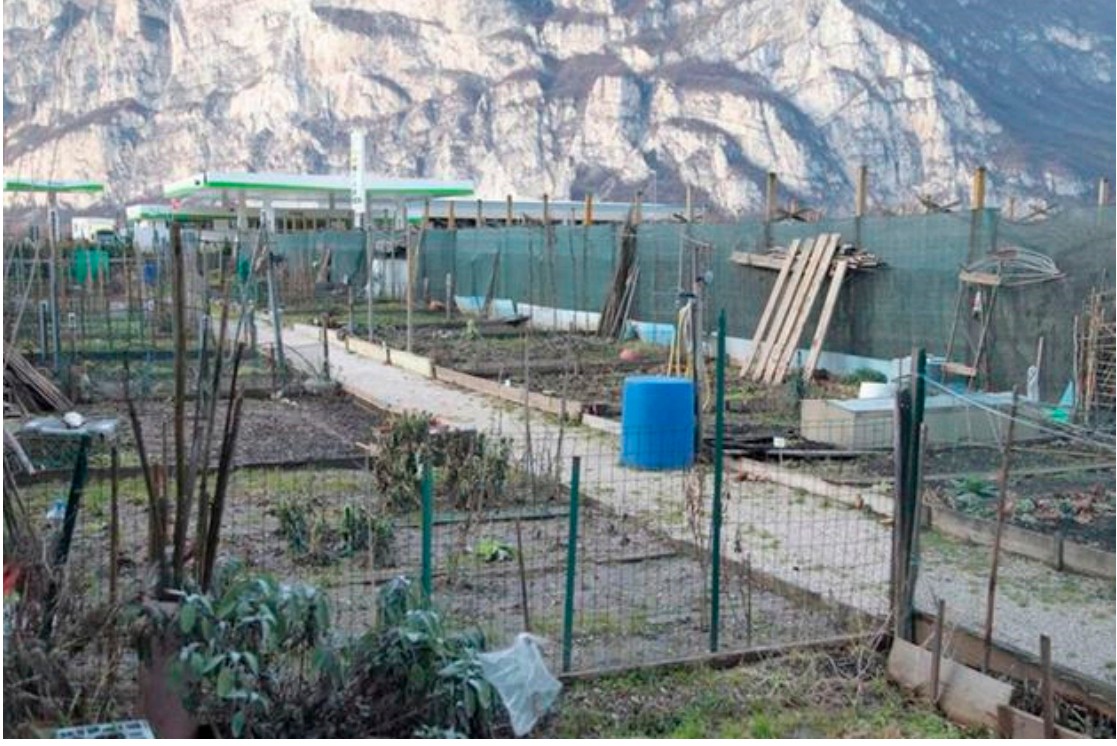

**Figure 5.** Some UGs in Salorno. Source: Municipality of Salorno, 2020.

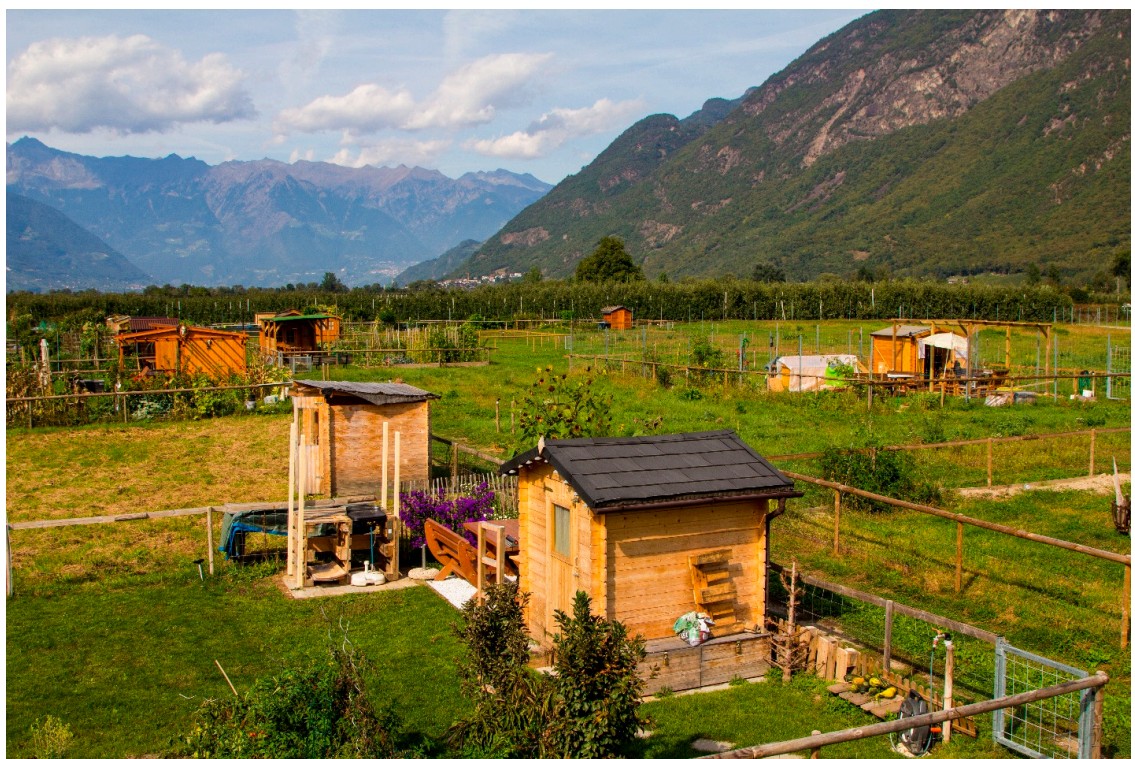

**Figure 6.** Some UGs in Terlano. Source: Othmar Hillebrand, 2020.

*6.2. The Different Meanings of these Experiences as Perceived by Municipal Administrations*

The five municipalities that promote UGs in their municipal areas assigned a score between 0 (not at all important) and 5 (very important) to all 14 meanings identified and grouped into four categories following the literature review.

Motivations related to social integration received the highest scores. The five municipalities consider opportunities for socialization and support for elderly people to be extremely important, and refer to strengthening family ties as an essential motivation underpinning their commitment to urban gardening. The first of these motivations is the most quoted in absolute terms, with all five municipalities attributing 5 points to it. The second scores well overall, though Vipiteno and Vadena did not rate it highly. The third motivation receives the same vote, though Vadena considers it less significant. Among the social considerations, two motivations—i.e., the social and ecological memory of ancient practices and strengthening personal and social identity—were least cited in absolute terms. This implies that mayors do not take them into account in their decisions. The last two social concerns—i.e., strengthening of social cohesion and participation and sharing values—are more controversial. The former motivation is ranked highly by mayors and municipal offices, particularly in Egna and Salorno. The latter, meanwhile, received the highest score in Egna, while all other municipalities gave it low marks.

These scores are also confirmed by the individual statements expressed by each interviewee. For example, municipality 1 wrote: "It was important for the municipal administration to create a place for socialization with regard to ecology". Municipality 2: "With this project we want to strengthen inclusion and socialization and offer new meeting places. I know that the majority of people have their backyard garden, but they cultivate it alone". The third mayor is even more specific when he underlines the social importance of UGs, claiming: "With this project (UGs project) we try to give each citizen the opportunity to cultivate a piece of land by socializing and exchanging opinions and ideas with other gardeners". Municipalities 4 and 5 highlight the particular importance of these projects to the elderly population, at risk of loneliness ("UGs are places where pensioners can spend some free

time and chat with other gardeners their own age"), and to families ("Cultivating UGs, families spend more time together).

Many of these mayors do not regard the strengthening of personal identity as a significant motivation "because this aspect concerns each individual person and I cannot speak for others" (Municipality 2). Similarly, they do not rank the memory of traditional practices highly "because the gardens are cultivated simply, rather with few tools, but with the same techniques of some years ago, not so many years ago" (Municipality 1). Mayors responded similarly with reference to the "sharing values" motivation: "Values concern each individual, therefore I cannot answer for my citizens" (Municipality 2) or "Values are difficult to identify, as well as to standardize and disseminate through cultivation" (Municipality 1).

Other clusters of motivations are also subject to significant debate, including the health effects. Within this category, mayors attribute great importance to the role of urban gardening in improving eating habits, while assigning little relevance to citizens' psycho-physical wellbeing. This is true for Vipiteno. The mayors/municipal offices of Bolzano, Egna, and Vadena, on the contrary, attribute less importance to food-related issues and more to health-related motivations. Their statements confirm their scores. Municipality 1 states that by "cultivating, people are encouraged to consume what grows in their garden and therefore more vegetables". This is contradicted by the mayor of Municipality 3, who notes that "urban gardens alone cannot change the eating habits of elderly people who have eaten the same things for decades", and the mayor of the town 4, who says that "many gardeners do not eat what they grow and give it away." In addition, Municipality 2 declares that "elderly people who cultivate a UG move around, they exercise", while another adds "getting your hands dirty, being surrounded by green relaxes the mind".

Alongside the social issues, motivations included in the urban requalification category are also highly ranked. In almost all municipalities, cleaning up vacant lots, reducing crime, or simply beautifying the neighborhood, protecting the landscape and the environment, and reducing pollution receive the highest scores. Of these, reducing pollution scores highest. This indicates that local municipalities regard pollution as a worrying phenomenon and UGs as a possible solution to reduce it. Bolzano attributes all of its highest scores to these concerns, underlining its interest in environmentally friendly and protective practices. Vipiteno assigns high scores to these motivations. On the contrary, these are less relevant to Vadena.

The mayors' and public offices' statements confirm these scores. Municipality 3 declares that "in my municipality, I want to preserve the rural landscape and offer a clean place". Mayor 1 adds that "during the day, there is a significant amount of pollution from normal travel and I want to implement some measures to reduce it and urban gardening may be one solution". Mayor 2 states that "my municipality is green, and I want to preserve the common space and I do not want vacant spaces to be able to be occupied by parking".

Little importance is attributed to the motivations related to the food security. Only the municipality of Bozen attributes a few points to this option. Mayor 4 explains: "In my municipality, elderly people cultivate gardens to pass the time, they have an income that allows them to shop, they do not need a garden to feed themselves". As such, UGs are not considered a laboratory for food self-sufficiency initiatives. This is confirmed by Municipality 5: "Gardeners regularly buy food in markets or supermarkets, they do not need a garden to feed themselves". Municipality 3 is open to some forms of supportive or gift economy: "Somebody grows flowers, somebody gives harvested vegetables".

All of the scores assigned by the mayors of the 5 municipalities interviewed are shown in the Figure 7.

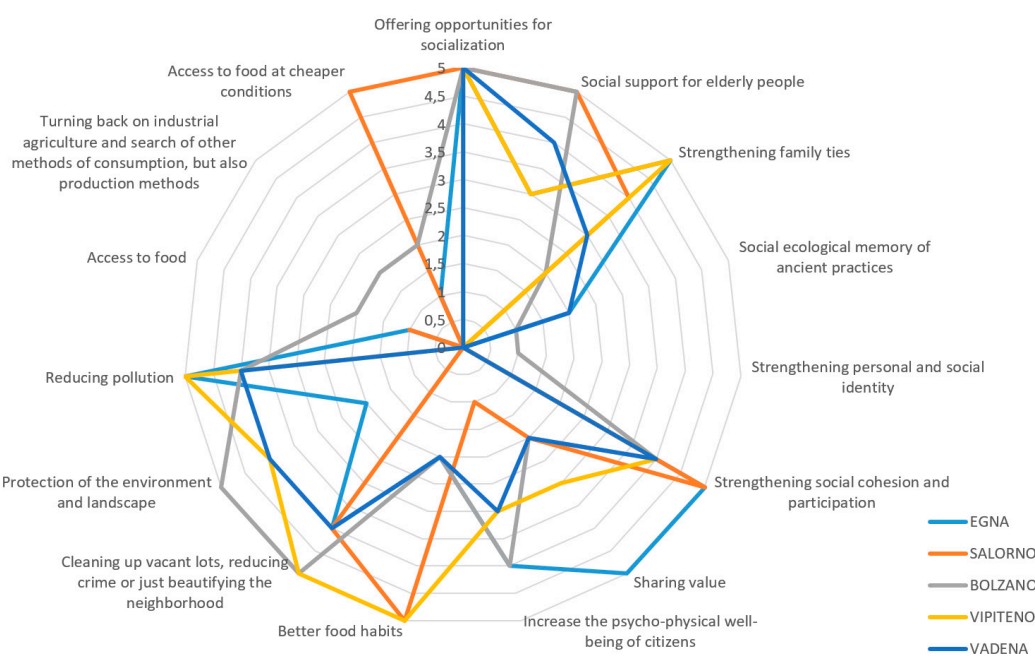

**Figure 7.** The motivations of urban gardens: The perspective of the municipalities. Source: Own elaboration based on direct interviews, 2019.

### 6.3. The Different Meanings of These Experiences as Perceived by Local Gardeners

16 gardeners were identified from across the five municipalities considered, and asked to respond to the second part of the text before submission to the municipalities. In other words, they were asked to freely state the motivations driving their decision to cultivate a UG, assigning a score to the motivations list. Figure 8 illustrates the average values of the scores given by the gardeners.

Motivations relating to social integration and urban requalification ranked highest. Gardeners regard the opportunity for socialization and support for elderly people as extremely important, and also consider the strengthening of family ties to be an essential motivation underpinning their commitment in urban gardening. The highest scores are attributed to these. The first of these motivations is the most quoted in absolute terms: On average, all gardeners across the five municipalities assign it the highest score. The second motivation concerning support for the elderly is highly ranked, though Vipiteno and Vadena attribute lower scores to it. The motivation linked to strengthening family ties receives an equal score, though Vadena considers it less relevant. Among the social concerns, two motivations—i.e., the social and ecological memory of ancient practices and strengthening of personal and social identity—are the least cited in absolute terms. They received zero points, implying that gardeners do not take them into account in their decisions. The last two social concerns—i.e., strengthening social cohesion and participation and sharing values—are more divisive. The former motivation is ranked highly by all the gardeners. The latter receives the top score among Egna gardeners, with all other gardeners assigning no or low points to it.

These scores are also confirmed by the individual statements expressed by the gardeners. For example, gardener 1 affirms that "I cultivate an urban garden so I do not feel lonely, to have fun, get out of the house, meet new people". Gardener 2 confirms: "I go to the garden, I have fun and I do not think about age-related problems, sometimes I meet some other gardeners, I exchange a few words with them, we discuss how much we cultivate". The third gardener is even more specific about the garden's role in strengthening family ties: "Sometimes I bring my nephew to the garden, he enjoys playing with the soil, sowing, weeding. While I see him having fun, I have fun too. In the meantime, I tell him the various names of the plants, when they are to be cultivated and harvested, and I like to introduce him to these things, because vegetables do not grow in the supermarket".

Many gardeners do not consider the strengthening of personal identity to be a significant motivation because "it is not by cultivating the garden that I strengthen my identity" (gardener 4) or "I am over 60 years old, I have my personality, it is too late to change it" (gardener 5) or "I perceive the identity of the place more by going to the local festivals or the old ways of cultivation still used in some mountain farms today" (gardener 6). Motivations concerning the strengthening of social cohesion and participation are even quoted among gardeners. This is demonstrated by some statements such as "Since I began cultivating the garden I feel more comfortable among the people" (gardener 3) or "Since I began cultivating the garden I am more in contact with the municipality and I am more informed of its activities and events" (gardener 6) or "I am part of a group, since I stopped working, I have not had the feeling of being part of a team" (gardener 4). Sharing values is not cited as a concern. This is because gardeners state that "with other gardeners I talk about the weather, the crops, but I do not have serious conversations" (gardener 7) or "my values belong to the private sphere of my life and I do not want to share them with anyone" (gardener 8) or "I cannot disseminate values through cultivation" (gardener 6), "that is the task of families and educators" (gardener 10).

Other motivations are largely divisive. This is the case with regard to the health effects, with gardeners attributing great importance to the role of urban gardening in improving eating habits, and little relevance to the psycho-physical wellbeing of citizens. This is true for Salorno and Vipiteno. Gardeners in Bolzano, Egna, and Vadena, on the contrary, attribute less importance to food-related issues and more to health motivations. Their words confirm the scores awarded. Gardener 1 states that "I want to grow a garden so I can eat fresh vegetables every day." He is contradicted by gardener 3 who declares ""UG alone cannot change my eating habits, I have eaten the same things for decades". One gardener in municipality 4 introduces the concept of UG cultivation as a potential driver of the gift economy: "I do not eat all of what I grow and give my friends and family all of what I produce". Gardeners 10 and 11, meanwhile, use similar wording to state that cultivating the garden allows them to exercise and keep moving and relaxing.

Together with social issues, motivations associated with the urban requalification category also rank highly. In almost all municipalities, cleaning up vacant lots, reducing crime, or simply beautifying the neighborhood, protecting the landscape and the environment, and reducing pollution receive the highest scores from gardeners. Of these, the motivation linked to reducing pollution ranks highest. This indicates that gardeners are aware that they live in communities where the natural environment is important, but is threatened by increasing human pressure, particularly from growing levels of pollution. As such, they state that their desire to cultivate a garden was in part motivated by seeking to ensure that the lands were not used for other activities, and to beautify the site or preserve the local landscape.

These concerns receive the highest scores from gardeners in Bolzano, underlining their interest in environmentally friendly and protective practices. Vipiteno and Vadena gardeners also ranked these motivations highly. For gardeners in Egna, meanwhile, these are less relevant.

These scores are confirmed by the gardeners' statements. Gardener 13 declares that "I would like to make the landscape in my municipality more beautiful and clearer". Gardener 14 adds that "there are more cars around, you can never find parking, the pollution increases, I do not want them to park there".

Little importance is attributed to motivations concerning access to food and research into new production and consumption methods as alternatives to industrial production. Only gardeners in Egna and Vipiteno attributed a few points to these considerations. Gardener 15 tries to explain this: "I go shopping in a supermarket or village shop, sometimes directly from the producers". Gardeners 16, 1, 4, and 5 raise the same concepts. Just one gardener states that "for me, my garden is a laboratory of food self-sufficiency, I try to cultivate what I need to feed my family as I do not want to be dependent on supermarkets". On the contrary, the more strictly economic motivations emerge as more significant. Gardeners place great importance on the opportunity to save money. "Cultivating a garden saves on vegetable costs", according to gardeners 2, 3, 6, 7, and 9.

All scores given by the gardeners from the 5 municipalities interviewed are shown in Figure 8.

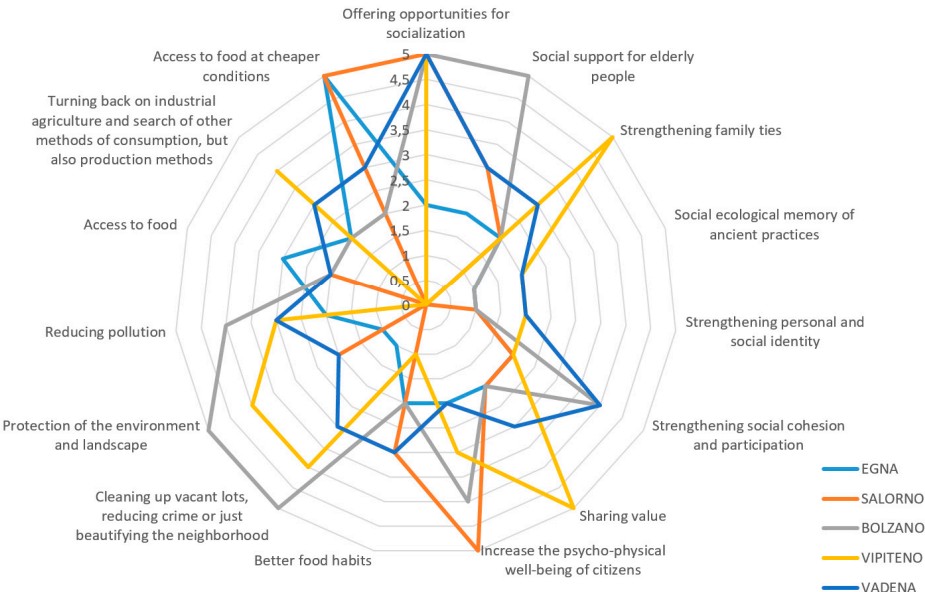

**Figure 8.** The motivations of UG: The gardeners' perspective. Source: Own elaboration based on direct interviews, 2019.

*6.4. The Different Meanings of these Experiences as Perceived by the Local Population Not Directly Involved in Urban Gardening Initiatives*

In the 5 municipalities considered, 20 people not directly involved in urban gardening initiatives (10 of whom are elderly, and 10 who represent families) were asked to express their ideas and perceptions of UGs based on the same list of motivations scored by mayors and gardeners. They expected to vote these motivations and add their thinking about the role of UGs. Figures 9 and 10 illustrate the average scores assigned by the elderly respondents and families, respectively.

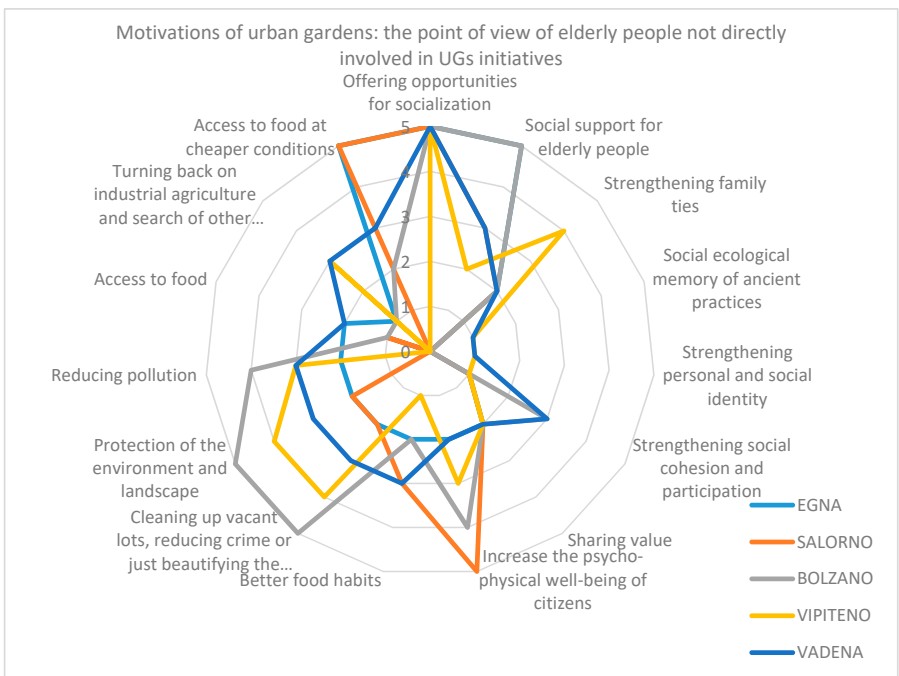

**Figure 9.** The motivations of UG: The perspective of elderly people not directly involved in UG initiatives.



Three out of the ten elderly respondents had no knowledge of the UG initiatives in their municipalities. Following a brief explanation, they expressed their ideas in the same way as the other seven respondents. Once again, social integration and urban requalification emerged as the most important motivations. This group regards the opportunity for socialization and support for elderly people as extremely important, but place less significance of the role of the gardens in strengthening family ties. They place no importance on their contribution to protecting the social and ecological memory of ancient practices and strengthening personal and social identity. This is also true of strengthening social cohesion and participation and sharing values, and will receive little consideration.

These scores are also confirmed by the individual statements expressed by each person. For example, person 1 states that "I think that urban gardens offer an occasion to meet other people". Person 2 confirms that "if I cultivated an urban garden, I would have fun. I would not feel lonely". The third respondent is even more specific about the role of the garden in strengthening family ties: "I do not think that this initiative could help family to spend more time together. Job, children, houses.. too busy, it's hard to meet on Sundays or holidays, let alone in a garden". In terms of values, person 3 explains: "I do not think UG helps share values, each of us already has his personality and a system of values in which we believe". Differences among municipalities are not particularly evident, on average municipalities assign more or less the same scores. Health effects are broadly divisive. Respondents attribute less importance to the role of urban gardening in improving eating habits and great significance to the psycho-physical wellbeing of citizens. This is true for Salorno, Bolzano, and Vipiteno; this is not true for the remaining municipalities. Their statements confirm the scores awarded. Person 1 states that "I would like to grow a garden so I can eat fresh vegetables every day." He is contradicted by person 3, who declares ""UG alone cannot change my eating habits, I have eaten the same things for decades". Person 6 states that "thanks to cultivation, I think we old people can keep in good health", noting the positive role that this might play in combating a sedentary lifestyle.

Together with social issues, motivations included in the urban requalification category also ranked highly. In almost all municipalities, cleaning up vacant lots, reducing crime, or simply beautifying the neighborhood, and protecting the landscape and the environment received the highest scores. Unlike other respondents, older people not involved in urban gardening do not believe that this initiative might reduce pollution. This indicates that these people are aware that they live in communities where the natural environment is important, but are not aware of the growing risk of pollution. This observation does not apply to those living in Bolzano, probably because they are more aware of the problem of pollution as they live in an urban area. By assigning high scores to this motivation, Bolzano residents express their interest in environmentally friendly and protective practices. Vipiteno and Vadena gardeners also ranked these motivations highly. They are of less significance, on the contrary, to respondents from Egna.

The statements made by the relevant respondents confirm these scores. Older respondent 7 declares: "I live in a beautiful village and I do not want more parking in the vacant spaces". Older respondent 8 adds: "there are more cars around, you can never find parking, the pollution increases, I do not want them to park there".

Little importance is attributed to motivations related to access to food and research into new methods of production and consumption as alternatives to industrial production. Only non-gardeners in Egna and Vadena attribute a few points to these categories. Person 1 tries to explain this: "I go shopping in a supermarket, sometimes I go to the farms and I buy directly from there". Person 4 and person 5 raise the same concepts:" I have money, I can by food directly in the supermarket, I do not need to cultivate to feed myself". However, respondents believe that growing a garden can help people save money on food shopping. "Cultivating a garden could help to save on vegetable costs" the non-gardeners 2, 3, 6, and 7 stated.

Figure 10 presents the perspective of 10 people interviewed as representatives of their families.

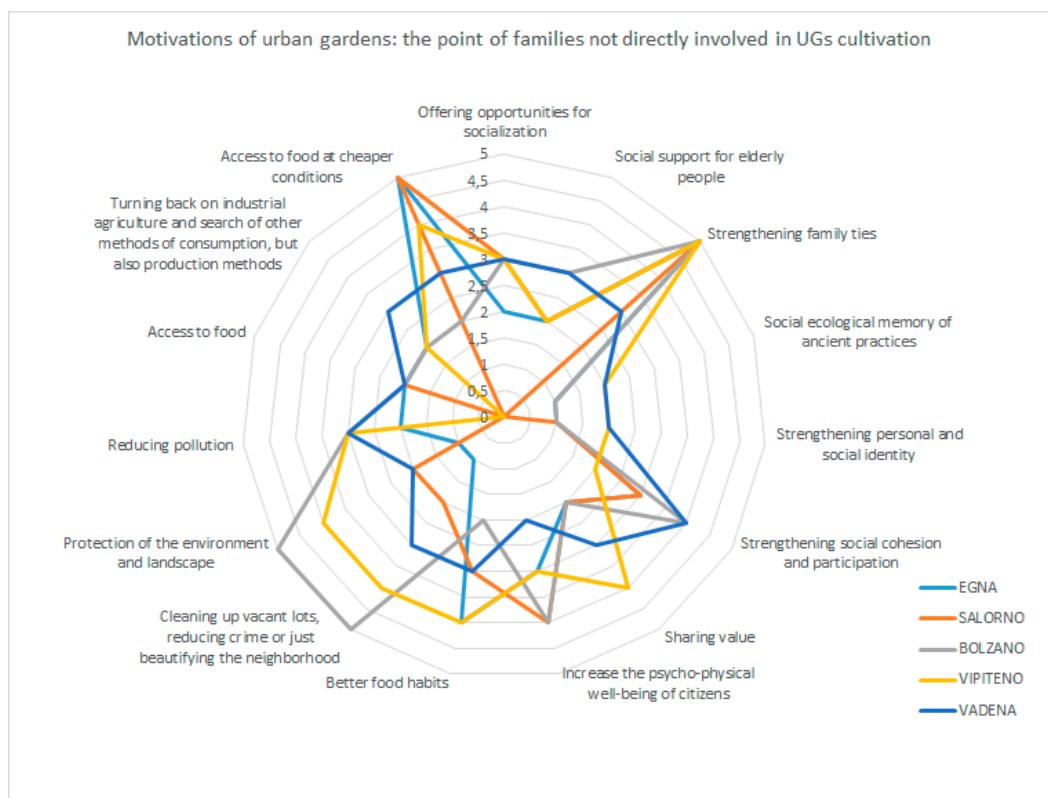

**Figure 10.** The motivations of UG: The perspective of families not directly involved in cultivating UGs.

Four of ten of the families had no knowledge of the UG initiatives in their municipalities. Following a brief explanation, they expressed their ideas in the same way as the other six respondents. For this group, too, social integration and urban requalification are the most important themes. These people place less importance on the opportunity to support elderly people, while showing a great interest in socializing in general and strengthening family ties. They place little importance on other social issues, and specifically the potential of UGs to contribute to preserving the social and ecological memory of ancient practices and strengthening personal and social identity. These scores are also confirmed by the individual statements expressed by each person. For example, person 3 affirms: "Through cultivation, I think I would meet other people". Person 6 confirms that "If I cultivated an urban garden I could have new relationships and increase my contacts". The fourth respondent is even more convinced of the importance of gardens in strengthening family ties: "I can spend more time in the open air with my children". More so than the elderly people, these respondents regard UGs as having an important role in terms of strengthening social cohesion and participation as "experiences of sharing and for the creation of networks, cohesion, systems of relationships which help people in case of difficulties" (person 4). Alongside considerations of the health effects, this latter effect is strongly divisive. Respondents from Bolzano and Vipiteno attribute higher scores, while those in the remaining municipalities are more uncertain.

Together with social issues, motivations in the urban requalification category rank highly, but differences are evident. In more urbanized municipalities like Bolzano and Vipiteno, respondents regard UGs as useful in terms of cleaning up vacant lots, reducing crime, or simply beautifying the neighborhood. Unlike other respondents, however, those based in Egna and Salorno and not involved in urban gardening do not believe that the initiative can reduce pollution. This indicates that these people are aware that they live in communities where the natural environment is important, but not a risk of growing pollution. The statements of family representatives confirm these scores. Respondent 5 declares: "Near my home, the landscape is beautiful and I do not want new houses to be build. No, I do not want more parking near my home".

Less importance is attributed to motivations concerning access to food and research into new methods of production and consumption as alternatives to industrial production. Only the non-gardeners in Bolzano and Vadena attributed a few points to these considerations. Person 6 tries to explain this: "Fortunately, I have a job and money to go shopping in a supermarket, when I can I go directly to farmers or farmers´ markets and I buy food from them". Person 9 and person 10 explain the same concepts: "Cultivating is stressful and I can only grow some vegetables. In a supermarket I can find any product, not just season items" or "I can by food directly in the supermarket, I do not need to cultivate to feed myself". However, all respondents believe that growing a garden can save money on food shopping. "Cultivating a garden could help to save on vegetable costs" the non-gardeners 2, 4, and 6 stated.

## 7. Discussion

The findings of this research suggest that, despite a growing recognition of the role of urban gardens in addressing social and environmental pressures, they are a relatively scarce commodity in South Tyrol. UGs are present in just eight out of the 66 municipalities that responded and are all located at low altitudes. No UGs are cultivated above 1000 m, though most municipalities in this province are at higher altitudes.

Most of the UGs analyzed are cultivated on municipal land, and as such are community gardens. In three out of the eight municipalities considered, gardens are cultivated on private land: This goes against the trend in the rest of Italy, where municipal management prevails. This probably arises from a "shadow" demand for UGs that it is not entirely satisfied by local municipalities, due to the limited number of available vacant spaces. Consequently, aspiring gardeners decide to lend private lands for cultivation. This suggests that local administrations should revise their future UG plans, to consider allocating additional municipal lands for gardening, and support the rezoning of private lands through municipal plans revisions. Municipalities cannot oversee private agreements between private owners and aspiring gardeners, and contractual conditions between these parties can differ from one municipality to the next. This form of agreement is not considered in this study, and may be the subject of future investigations.

Gardeners are male, older, and from South Tyrol. The consideration given to families in Vadena is an exception in South Tyrol and is not a frequent occurrence. Few other municipalities consider applications to cultivate a UG from those under 55 years of age. This demonstrates the willingness of the aforementioned municipality to extend the initiative to other age groups, including families and younger people.

Regarding their motivations, municipalities state that they established UGs for social reasons. They regard increasing socialization and social support for older people, as well as strengthening family ties, as essential motivations underpinning their commitment to urban gardening. Their attitudes confirm the assumptions by Mudu and Marini (2018) and Hawkins et al. (2013) that point to the role of UGs in promoting social integration, and a desire to transform local society into a more inclusive community. This is interesting in academic terms, as it confirms the assumptions already validated for urban areas, but it is also valuable from a social perspective, as it demonstrates a desire to cement the links between individuals with a view to rebuilding the network of relationships that served as a support system in mountain areas, effectively compensating for a lack of state supports. However, this desire is at odds with a lack of consideration of the role of UGs in terms of expressing personal and social identity. Motivations such as social and ecological memory, and personal and social identity rank lowest. This is also interesting according to the same points of view. Academically, this lack of consideration for these factors contradicts Clayton (2007) and Crompton and Kasser (2009) who, on the contrary, note the importance of UGs in this sense. Socially, little consideration is undesirable as it does not lead to the formation of local social capital, as theorized by Veen et al. (2016), which risks falling into decline due to depopulation.

Municipalities regard UGs as essential to improving environmental conditions. This is confirmed by the high scores assigned to certain motivations such as beautifying the neighborhood, reducing pollution, and preserving rural landscapes. The scores attributed by municipalities to these aspects confirm their concerns about landscape quality and attempts to preserve them by realizing UGs. Their concerns and judgments are in line with those expressed by municipalities in urban areas. Attributing less importance to food issues, they recognize the self-autonomy of the local population in terms of food, and do not regard UGs as an important initiative to support food supply. This may be because they not consider their municipalities to be at risk of food desertification. Municipalities' opinions on the importance of UGs in terms of health matters differ greatly. While recognizing the relevance of UGs in this regard, they prefer not to draw conclusions about the contribution of gardens, something that partly contradicts the literature.

Gardeners assert that social and environmental reasons are the most important. Among the social concerns, gardeners regard the opportunity for socialization and social support for older people as extremely important, and see strengthening family ties as an essential motivation underpinning their commitment to urban gardening. In this way, they point to the social and integration functions of the garden, and to its importance as a remedy for loneliness and social exclusion. Gardeners consider the recovery of social and ecological memory of ancient practices and the strengthening of personal and social identity to be less important, while the final two social concerns—i.e., strengthening social cohesion and participation and sharing values—are more divisive. Indeed, some interviewees regard these two aspects as important precisely because they highlight the decline in the social context in the area where they live and the opportunities offered by the UGs to restore it. In their opinion, gardens serve as an interesting meeting place to connect with other people, but not to share values or experiences. Gardens also help to strengthen family ties because, through cultivation, grandparents spend more time with their grandchildren. Their opinions on health effects demonstrate the great importance attributed to the role of urban gardening in improving eating habits. One of the gardeners even links UGs to the gift economy, i.e., generating generosity circuits and offering the opportunity to give part of the produce to friends and relatives. On the contrary, they indicate the low significance attributed to the role of the gardens in promoting citizens' psycho-physical wellbeing. Together with social issues, motivations included in the urban requalification category also rank highly. Gardeners aim to reduce local pollution through cultivation, preserving the local environment and common spaces. This is in keeping with the opinions expressed by the municipalities in these regions, and with the findings in urban areas. Little importance is attributed to motivations related to access to food and research into new production and consumption methods as alternatives to industrial production. On the contrary, the high scores assigned to the opportunity to save money confirm gardeners' willingness to cultivate UGs for cheaper vegetables. Part of the people not directly involved in UGs initiatives do not know anything about the initiatives implemented in their territories. Following an explanation and in keeping with those with prior awareness of the practices, they partially confirmed the statements made by the gardeners, emphasizing the role of UGs in socialization for older people (this emerges as a particular priority among older respondents), in strengthening family ties (families), and in promoting landscape preservation and reducing food costs.

## 8. Conclusions

This paper investigates the spatial distribution of urban gardens and the motivations underpinning their cultivation in mountain areas such as South Tyrol. This is the novelty of this paper because previous surveys have focused on UGs in urban and metropolitan areas, without reference to other contexts such as mountain or rural areas. A further element of novelty is represented by the fact that the article presents the opinions of several categories of actors involved in gardening projects. As well as the perspective of gardeners, which is typically considered in the literature, this article also analyzes the attitudes of municipalities and municipal offices, as well as those of older people and families not directly involved in such projects. The decision to include and discuss different perspectives is

surely an element of innovativeness that can contribute to enriching the current debate, while at the same time offering material for discussion. Despite growing interest, no organizations have been established in the region of the sort that emerged in France and Germany during the 20th century to promote UG projects. This serves as an obstacle to the sharing of ideas and projects, and a greater diffusion of these practices. This study provides municipalities with the opportunity to reflect on their decision to promote UGs and better reformulate their territorial policies to meet the real needs of current and potential gardeners. On the other hand, it gives gardeners an insight into the perspective of the municipalities, helping them understand their position. The interviews with local gardeners and administrators reveal that the experience is judged positively by both sides, especially as regards the social and environmental implications. Food issues are considered to be of little relevance, except for the savings in terms of food costs that can be achieved from direct cultivation. It may prove interesting to revisit these opinions in light of the food supply issues arising from the restrictions on personal mobility put in place to contain the spread of Covid-19. It is possible that theses issues may assume more significance to the interviewee in light of this situation.

The attention reserved by the interviewed is positive for South Tyrol, which is currently subject to worrying demographic and environmental pressures. UGs are in fact regarded as a potential solution to mitigate the negative effects of loneliness among older people and the depletion of social and natural capital, but are deemed less significant as a solution to improve food access.

Future surveys should extend the sample of the municipalities, gardeners, and people not directly involved in UG initiatives interviewed, analyzing the spread of the phenomenon in remote alpine and rural areas in other Italian and European regions.

**Funding:** The author thanks the Department of Innovation, Research and University of the Autonomous Province of Bozen/Bolzano for covering the Open Access publication costs.

**Acknowledgments:** The author thanks also prof. Lina Napolitano for her suggestions.

**Conflicts of Interest:** The author declares no conflict of interest.

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
