# Peer review of "The Motivation of Urban Gardens in Mountain Areas. The Case of South Tyrol"

_sustainability, doi:10.3390/su12104304_

Round 1
Reviewer 1 Report
A brief summary:
This papaer presents a subject rarely studied but with great importance now more than ever, given the excessive urbanization and globalization of the world we live in. The exposure of such a form of human interaction and leisure, combined with the rediscovery of one of man's ancestral occupations in a scientific work, is to be commended; therefore we wish to congratulate the author for his work. This subject is certainly within the scope of the sustainability of both the territory and the resident population.
As far as the manuscript is concerned, it contains a great deal of information and presents at length the subject matter and how the UGs are perceived in South Tyrol. For the improvement of the manuscript I think that more attention should be paid to graphic and cartographic representations, and where written information can be replaced with more expressive visuals, this should be done. At the same time, it is advisable to give up information and descriptions that are not directly related to the present study. The questionnaire is preferable to be applied to a large number of respondents belonging to different social categories, in order to have a better overview of what the UGs represent, not only for the decision makers in the regions concerned, but also for other sectors of the area population.
Broad comments:
Chapter 1 – Introduction - It would be necessary for this introductory chapter to find a detailed definition of UGs, which means their purpose and importance, as well as the difference between traditional gardens and UGs. Also a brief history of them would be very interesting.
Chapters 2 and 3 - The chapters are very well argued, the quotations that refer to the specialized literature are well organized and capture the essence of the works.
Chapter 4 – Methods - All statements related to the interpretation of data should be removed, therefore they should be introduced in the chapter dedicated to results or discussions, to the current chapter remaining only statements about the modalities of data collection and interpretation. At the same time, it should be better explained how the data were further analyzed, as well as the softwares used for the analysis and graphical and cartographic representation of the data.
Chapter 5 – Test are: the South Tyrol - There is a lot of information that has no connection or any relation of interdependence with what UG means, both in general and in the studied area. The exact positioning, as well as the provinces, cities and municipalities and natural landscape could be presented much easier and expressively through a detailed map on a DEM. Such a representation could better capture what South Tyrol means, without much numerical data not related to the purpose of this study.
Chapter 6 – The urban gardens in South Tyrol - Although the text describing satisfactorily the role and importance of the UGs in South Tyrol, as well as the motivations that compel the inhabitants to go to such places to spend their free time, in the study I did not find any figure that would show such a place. Photographs showing what a UG in South Tyrol means must be introduced, this will undoubtedly raising the value of the work. It would be more interesting to present gardens with different particularities from the study area.
Chapter 4 and 6 - Given that the main scoup of the UGs is to serve the population, it would be necessary to find within this paper the opinions of ordinary people about the importance of gardens in society (whether or not the respondents frequent the gardens).
Specific comments:
Rows 99-107 - I don't think this phrase is needed. instead, it could detail the purpose of the present study, what it wants to highlight and what results are expected, without a list of paragraphs.
Rows 109-119 - These lines describe in great detail what UG represents. It would be more appropriate if this description were to be included in the introductory chapter.
Rows 208-232 - Perhaps here it should be explained with what the UG is differentiated from the other forms of agriculture practiced by the population from the countryside, but also what might determine them to choose UG next to or to the detriment of the agriculture they already practice.
Rows 365-368 - Global warming is a rather demanding term and difficult to use unless it is accompanied by data or reference works in the field. But even so it is hard to say that the area is affected by climate change or global warming. It may be best to remove these lines from the outset, since they do not have overwhelming importance for the study.
Figure 1 - Positioning information at the level of Europe and Italy and also about the natural or relief framework should be presented in another map dedicated only to a better understanding of the study area. The present map may be devoted only to the quantitative presentation of the data on the spatial distribution and the number of UGs resulting from the questionnaires applied or following the discussions with the mayors or the representatives of the territorial administrative units concerned. In terms of the content of these map: the writing of the legend of the map is too small and difficult to read, it would also be advisable to enter the scale of the map and indicate the north, it would also be more appropriate to facilitate the reading to use different colors to express number of UGs in each province (Ex: red - 162 UGs, yellow - 20 - 30 UGs, green - 10 - 20 UGs, pink - under 10 UGs, gray - none, white - NA).
Reviewer 2 Report
I would like to congratulate the author on a very interesting piece of research. However, the following issues need to be addressed, in order for the paper to be accepted.
General comments:
The paper requires an extensive proofreading and editing. The writing style is informal in certain points. Please refrain from writing in first person: lines 78, 271, 275, 288 etc.
Introduction
Lines 84-87: Novelty not clearly expressed
Lines 88 needs to be reviewed in terms of English. In addition, a paper would normally refer to aim rather than objectives
Lines 90: check English – subjects and verbs should match in number and persons.
Line 95 – it seems that the author is performing just one interview
Line 97 – Semi-structured… what???
Section 3
Lines 233 to 240 – The author seems to make several assumptions in a supposedly Literature Review section not supported by any evidence at this stage in the paper.
Methods
The method section requires clarifications.
The author refers to interviews in the introduction and to questionnaire in methods section. This are two different research methods that requires different approaches in terms of sampling, data collection and analysis. The author needs to be clear and consistent with the language.
The questionnaire seems to incorporate open-end questions and ordinal questions. The author needs to be more specific on the approaches used to analyse these questions
Line 300 – Gardeners…what?
Lines 305-306: “Given the low number, in fact, the answers of these gardeners were easily traceable to their thought and therefore it was not correct to rework them”. This statement needs to be clarified and modified as the author need to assure anonymity of the research participants; and addition the authors should present the performed analysis.
Lines 311-312: The author mentions statistical significance. However, as this seems mainly a qualitative type of research, this should not be considered.
Section 5 should be a subsection of section 4
Section 6 seems a mapping exercise and it how this was performed should be better explained in the methods section
Discussions should not be a meagre repetition of the analysis, but it should present a meaningful and critical discussion.
Conclusions need to be strengthen.
Round 2
Reviewer 1 Report
I no longer have any specific suggestions for authors. I appreciate the work done and the dedication and also the fact that they implemented all the comments from the first review.
Author Response
thank you
Reviewer 2 Report
Dear author,
thank you for addressing the comments. The papers is very much improved. Please see below some additional comments for improvement:
As Table 1 is a summary from the literature, it should have an additional column where the refences for each row are provided
On lines 345-346 the author should explain the type of content analysis performed rather that the software tool used. Please see for example, Hsieh, 2005 https://www.ncbi.nlm.nih.gov/pubmed/16204405
Table 2 could easily fit in one page if resented in portrait rather than landscape, which would improve the readability.
Author Response
I reformulate tables 1 and 2 and define better the method